# Effects of exogenous EBR on the physiology of cold resistance and the expression of the *VcCBF3* gene in blueberries during low-temperature stress

**Jiahe Fan**[ID], **Xuedong Tang***, **Jiaxin Cai**[○], **Ruiyang Tan**[○], **Xue Gao**[○]

Department of Horticulture, College of Horticulture, Jilin Agricultural University, City of Changchun, Province of Jilin, China

[○] These authors contributed equally to this work.

* tangxd94@126.com

**Data Availability Statement:** All relevant data are within the paper and its Supporting Information files.

## Abstract

The northern highbush blueberry variety 'Duke' was used as the test material, and different concentrations of 2,4-Epibrassinolide (EBR) (0, 0.2, 0.4, 0.6, and 0.8 mg·L⁻¹) were applied during the bud expansion stage, with a second application administered at one-day intervals following the first. Samples were collected at the bud, flower, and fruit stages and subsequently treated with artificial low temperatures (2°C) after sampling. The effects of various concentrations of exogenous EBR on the physiological indices of cold resistance and the expression of the cold resistance gene *VcCBF3* in blueberry buds, flowers, and young fruits were investigated through comprehensive evaluation and correlation analysis. The objective was to identify the optimal concentration of EBR to enhance the cold resistance of blueberries. The results indicate that: (1) Under low temperature stress, the contents of soluble sugar, soluble protein and proline increased, along with the activities of superoxide dismutase, peroxidase, and catalase. The expression of the *VcCBF3* gene expression and the ascorbate-glutathione cycling system were up-regulated, and with the increase of EBR concentration, the expression of the *VcCBF3* gene initially rose and then declined. The content of malondialdehyde and the production rate of superoxide anion radicals decreased, and with the increase of EBR concentration, the content of malondialdehyde first decreased and then increased. (2) Overall low temperature resistance, flowers > buds > young fruits. (3) Appropriate concentrations of exogenous EBR can effectively mitigate freezing damage in blueberries caused by low temperatures. A comprehensive evaluation and correlation analysis of each cold tolerance index and the expression of the *VcCBF3* gene revealed that a treatment concentration of 0.4 mg·L⁻¹ had the most significant mitigating effect among the sprayed EBR concentrations of 0, 0.2, 0.4, 0.6, and 0.8 mg·L⁻¹.

**Funding:** The author(s) received no specific funding for this work.

**Competing interests:** The authors declare no conflict of interest.

## Introduction

Blueberry, which belong to the rhododendron family (Ericaceae) and are classified under the genus *Vaccinium*, are perennial, shallow-rooted shrubs. They produce nutritious fruits that are both sweet and tart, accompanied by a delightful aroma [1–3]. Changchun, located in northeastern China within an alpine region, experiences extended periods of low temperatures. In spring, as fruit trees begin to bud and flower, the flower organs exhibit a limited ability to withstand external damage. Temperature fluctuations, characterized by warming followed by cooling, can lead to occur 'inverted spring cold' phenomenon, resulting in the drying and wilting of flower buds, browning and dropping of flower organs, and even deformation of young fruits. Such conditions significantly hinder the development of the blueberry industry in the Changchun area [4]. In a comparative study examining the cold resistance of various lingonberry varieties, Ren B concluded that the soluble sugar (SS) content increased as the temperature decreased. The soluble protein (SP) content exhibited an initial increase, followed by a subsequent decrease. Although the free proline (Pro) content increased, it gradually declined with the intensification of stress and the extension of the treatment duration. Furthermore, the malondialdehyde (MDA) content initially rose before slowly declining [5]. Wei X et al. discovered that the MDA content in various blueberry branch varieties gradually increased as the temperature decreased, following an 'S'-shaped curve. In contrast, the activities of peroxidase (POD), superoxide dismutase (SOD), and catalase (CAT) exhibited a unimodal trend, initially increasing before subsequently decreasing [6]. Liu BH et al. observed that the ascorbate-glutathione (AsA-GSH) system demonstrated a pattern of increase followed by a decrease as the temperature declined from the pre-dormant stage to the post-dormant stage [7].

In recent years, exogenous substances have attracted considerable attention for their role in enhancing plant cold resistance. 2,4-Epibrassinolide (EBR) is a novel class of highly effective, broad-spectrum, and environmentally friendly plant growth regulators. It has the capacity to regulate plant growth and development, improve resistance to low temperatures, and is recognized as the sixth major plant hormone on the global stage [8–10]. EBR enhances the cold tolerance of plants by reducing the extent of membrane lipid peroxidation. It achieves this by increasing antioxidant capacity, lowering MDA levels, and decreasing the degree of lipid peroxidation in cellular membranes. Consequently, EBR improves the water-holding capacity of cells and tissues, thereby mitigating damage caused by low temperatures. Additionally, EBR modulates plant metabolism by inhibiting the generation of excessive free radicals and promoting the production of free radical scavengers, which stabilizes the structure and function of cellular membranes. EBR treatment also activates the antioxidant enzyme system, enhancing the activity of key antioxidant enzymes such as SOD, POD, and CAT. Furthermore, it induces the accumulation of non-enzymatic antioxidants, including glutathione and proline, which effectively scavenge reactive oxygen species (ROS) [11–13]. It has been demonstrated that EBR can enhance the cold resistance of various plants, including tea trees [14], peanuts [15], bananas [16], loquats [17], and others, playing a crucial role in mitigating low-temperature stress. Guo XL et al. reported that EBR improved the cold resistance of grape leaves [18]. Similarly, Yi CX et al. found that a 0.5 μM EBR treatment significantly increased the cold resistance of lychee [19]. Additionally, Hu SQ et al. discovered that post-harvest EBR treatment enhanced the cold resistance of peach fruits [20]. Currently, there are no reports on the effects of EBR on the cold resistance of blueberries under low temperature stress in China.

When plants are exposed to low temperature stress, they first detect the signals associated with reduced temperatures and subsequently activate transcription factors through a series of signal transduction pathways. These transcription factors bind to cis-acting elements, resulting in the expression of various low-temperature-responsive genes that produce numerous

functional proteins, thereby enhancing the plants' cold tolerance [21, 22]. The transduction pathway for low temperature signaling in plants has been extensively studied, with one of the most prominent pathways being the *CBF*- dependent pathway (ICE-CBF-COR) [23]. *CBF* transcription factors are part of the AP2/EREBP family, which plays a crucial role in the *CBF* dependent pathway and is conserved across a wide range of higher plants [24]. Current research indicates that *CBFs* are regulated by *ICE1*, *ICE2*, and the calmodulin-binding transcriptional activator CAMTA3 [25]. Notably, the transcription factor family recognizes cold-responsive and dehydration-responsive elements (CRT/DRE) in the *COR* promoter region, which is why *CBF* transcription factors are also referred to as dehydration-responsive element-binding factors (DREB) [26]. In Arabidopsis thaliana, the *CBF* gene, which is part of the DREB subfamily, comprises three copies: *CBF1*, *CBF2*, and *CBF3*. All three copies can be regulated by the upstream transcription factor *ICE*. The activation of *CBF* results in the upregulation of its downstream *COR* genes, thereby enhancing the cold resistance of the plants [25]. *CBF1*, *CBF2*, and *CBF3* are rapidly and transiently upregulated in response to low temperature stress, typically reaching peak expression levels after exposure to 4°C for 1 to 3 hours [27]. *CBF* can bind to downstream *COR* genes, and the proteins encoded by these *COR* genes can activate the expression of additional genes that encode osmotic regulators, either directly or indirectly. This process can increase the levels of soluble sugars and antifreeze proteins, repair membrane lipid damage caused by low temperatures, and ultimately improve the cold tolerance of the plants [26, 28]. Chinnusamy V et al. discovered that *ICE1* specifically binds to the Avian myelocytomatosis virus (MYC) cis-acting element in the promoter region of the *CBF3* gene. Furthermore, the overexpression of the *ICE1* gene resulted in increased expression of the *CBF3* gene and its downstream low-temperature-responsive genes in transgenic Arabidopsis thaliana plants subjected to low temperature stress. This enhancement significantly improved the cold tolerance of the transgenic plants [29].

In this experiment, we utilized the northern highbush blueberry variety 'Duke' as the test material to investigate the effects of various concentrations of EBR on several physiological indicators of cold resistance in blueberry buds, flowers, and young fruits. Additionally, we examined the expression of the key gene associated with cold resistance in blueberries, *VcCBF3*, under low temperature stress. The optimal concentration of exogenous EBR was identified to enhance the cold resistance of blueberries. This research aims to provide a cost-effective technology and methodology for improving cold resistance in blueberries during the flowering and young fruit stages. Furthermore, it seeks to establish a theoretical foundation for future studies on EBR's potential to mitigate damage caused by low temperature stress and to enhance the cold resistance of blueberries.

## Materials and methods

### Test materials

Ten-year-old 'Duke' northern highbush blueberry bushes under the same management conditions, with a consistent flowering period and in good growth condition, were selected. The sampling location was the small berry orchard in the experimental site of Jilin Agricultural University (125°41′E, 43°82′N).

### Test methods

The experiment involved applying various concentrations of EBR during the flower bud expansion stage on April 21. The treatment concentrations included 0, 0.2, 0.4, 0.6, and 0.8 mg·$L^{-1}$, with a second application occurring one day after the initial spray, resulting in a total of two applications. Fresh water served as the control treatment. The experimental design

followed a randomized block design, with three plants per plot and three replications. All other management conditions remained consistent throughout the experiment. Flower buds, fully opened flowers, and young fruits were collected from the upper middle crown of each test tree in the four cardinal directions (east, west, south, and north) two weeks after flowering, corresponding to the bud, full flowering, and young fruiting stages, respectively, to create a mixed sample. Following collection, the samples underwent low temperature stress treatment by artificially simulating a low temperature environment. In this experiment, the treatment temperature was set to 2˚C [30], a condition known to induce low temperature cold damage during the flowering and fruiting periods of blueberries. The cooling rate was maintained at 4˚C per hour. To simulate short-term low temperature events that occur in the natural environment and to observe the stress response of blueberries to this relatively brief exposure, the temperature was held at the target level for 8 hours. Following this period, the temperature was gradually increased to room temperature at the same rate. Once room temperature was reached, the samples were removed and immediately placed in liquid nitrogen for treatment. Subsequently, they were stored in an ultra-low temperature refrigerator for the determination of physiological indices and the expression of the cold-resistant gene *VCCBF3*.

## Project determination

**Determination of soluble sugar content.** SS content was determined using the anthrone colorimetric method [31]. A sample of 0.5 g each of cut and mixed buds, flowers, and young fruits was placed into a test tube. Next, 15 mL of distilled water was added, and the mixture was boiled in a water bath for 20 min. After boiling, the mixture was removed and allowed to cool. The solution was then filtered into a 100 mL volumetric flask, and the residue was rinsed several times with distilled water before adjusting the volume to the calibration mark:

$$\text{SS content }(\%) = \frac{CV_T}{10^6 WV_1}$$

Where: C-amount of glucose found from the standard curve, μg;
$V_T$-total volume of sample extract, mL;
$V_1$-volume of sample solution taken for colour development, mL;
W-sample weight, g.

**Determination of soluble protein content.** SP content was determined using the Coomassie Brilliant Blue G-250 staining method [31]. A total of 0.5 g of each sample—comprising cut and mixed buds, flowers, and young fruits—was weighed and ground into a homogenate with 5 mL of distilled water. The mixture was then centrifuged at 3,000 r/min for 10 min. To establish a baseline, 1.0 mL of distilled water was combined with 5 mL of Coomassie Brilliant Blue reagent. Next, 1.0 mL of the sample extract was added to a test tube, followed by 5 mL of Coomassie Brilliant Blue reagent. The mixture was shaken thoroughly and allowed to stand for 2 min. The color was then compared at 595 nm to determine the absorbance, and the procedure was repeated three times. The results were calculated using the following formula:

$$\text{SP content }(\text{mg/g}) = \frac{CV_T}{1000V_S W_F}$$

Where: C-check standard curve value, μg;
$V_T$-total volume of extract, mL;
$W_F$-fresh weight of the sample, g;
$V_S$-volume of sample added during the determination, mL.

**Determination of proline content.** Pro content was determined using the ninhydrin color development method [31]. A total of 0.5 g each of cut mixed buds, flowers, and young fruits were placed into stoppered test tubes. To each tube, 5 mL of a 3% sulfosalicylic acid solution was added. The tubes were then sealed and subjected to a boiling water bath for 10 min to facilitate extraction. After filtration, 2 mL of distilled water (control) and the filtrate were placed in separate test tubes. Subsequently, 2 mL of glacial acetic acid and 2 mL of ninhydrin reagent were added to each tube, which were then sealed and heated for 30 min in a boiling water bath. After cooling, 5 mL of toluene was added to each tube, and the mixture was shaken thoroughly to enhance extraction. The toluene layer was then carefully transferred to a cuvette. The samples were protected from light and allowed to stand until completely stratified. The absorbance of each sample was measured at 520 nm using a spectrophotometer, with the distilled water group serving as a blank control. Each measurement was repeated three times. The results were calculated using the following formula:

$$\text{Pro content } (\mu g/g) = \frac{C \times V_T}{W \times V_1}$$

Where: C-mass of Pro concent from the standard curve, μg;
$V_T$-total volume of extract, mL;
$V_1$-volume of assay solution, mL;
W-sample mass, g.

**Determination of malondialdehyde content.** MDA content was determined using the thiobarbituric acid (TBA) method [31]. First, weigh 1 g of a mixture of cut buds, flowers, and young fruits. Add 2 mL of 5% trichloroacetic acid (TCA) along with a small amount of quartz sand, and grind the mixture until it is homogenized. Next, add 8 mL of TCA for further grinding. Centrifuge the homogenate at 4000 r/min for 10 minutes. Aspirate 2 mL of the supernatant (using 2 mL of distilled water as a control) and add 2 mL of a 0.6% TBA solution, shaking well to ensure thorough mixing. Place the test tube in a boiling water bath and boil for 10 min, starting from the moment small bubbles appear in the solution. After boiling, remove the test tube and allow it to cool. Centrifuge the cooled solution at 3000 r/min for 15 min. Collect the supernatant and use the control as a blank to measure the absorbance values at 532 nm, 600 nm, and 450 nm. Repeat the procedure three times. The results were calculated using the appropriate formula:

$$C(\mu mol/L) = 6.45 \left( A_{532} - A_{600} \right) - 0.56 A_{450}$$

$$\text{MDA content } (\mu mol/g) = \frac{C V_T V_1}{1000 V_2 W}$$

Where: C-MDA concent calculated according to Eq, μmol/L;
$V_T$-total volume of sample extract, mL;
$V_1$-total volume of sample extract and TBA solution reacted, mL;
$V_2$-volume of sample extract reacted with TBA, mL;
W-sample mass, g;
1000-factor to convert mL to L.

**Determination of the rate of superoxide anion radical production.** The rate of superoxide anion radical ($O^{2\cdot-}$) production was determined through hydroxylamine oxidation [31]. Weigh 5 g each of cut and mixed buds, flowers, and young fruits. Add 10 mL of 50 mmoL/L PBS (pH 7.8) and grind the mixture. Centrifuge at 5000 g for 10 minutes. Take 1 mL of the supernatant and add 0.9 mL of 50 mmoL/L PBS, along with 0.1 mL of hydroxylamine

hydrochloride (using PBS instead of the sample supernatant as a blank). After mixing, the sample was incubated in a water bath maintained at a constant temperature of 25°C for 30 min. Subsequently, 1 mL of the incubation solution was extracted, to which 1 mL of p-aminobenzenesulfonic acid and 1 mL of α-naphthylamine were added sequentially. The reaction was then allowed to proceed in the water bath at 25°C for an additional 20 min. Following this, 3 mL of n-butanol was introduced, and the n-butanol phase was collected for the determination of absorbance at 530 nm after thorough shaking. Based on the measured A530, the $NO_2^-$ standard curve was referenced to convert A530 to $[NO_2^-]$. The concentration of $[NO_2^-]$ was then multiplied by 2 to obtain $[O^{2\cdot-}]$. Using the reaction time of the sample and hydroxylamine (30 min), along with the weight of the sample, the rate of $O^{2\cdot-}$ production was calculated, with the experiment being repeated three times. The results were computed using the following equation:

The rate of $O^{2\cdot-}$ production$[\mu mol/(min\cdot g)] = \frac{CV_TV_1}{1000V_2tW}$

Where: C-$[NO_2^-] \times 2$ from the standard curve, $O^{2\cdot-}$ concentration, μmol/L;

$V_T$-total volume of sample extract, mL;

$V_1$-volume of the mixture after reaction with hydroxylamine hydrochloride, mL;

$V_2$-volume of sample extract removed from the sample extract reacted with hydroxylamine, mL;

t-time of reaction of the sample with hydroxylamine hydrochloride, min;

W-mass of the sample, g;

1000-convert the volume mL to L.

**Determination of superoxide dismutase activity.** SOD activity was assessed using the nitroblue tetrazolium (NBT) photochemical reduction method [31]. Weigh 0.5 g each of sheared and mixed buds, flowers, and young fruits in a pre-cooled mortar. Add 1 mL of pre-cooled phosphate buffer and grind the mixture into a homogenate while maintaining an ice bath. Subsequently, add buffer to achieve a final volume of 5 mL. Centrifuge the mixture at 10,000 r/min for 20 min at 4°C; the resulting supernatant will serve as the crude SOD extract. Prepare four 5 mL transparent tubes: two for measurement and two for control. Follow the Table 1 to add the appropriate solutions. After mixing, place one control tube covered with a double-layer black paper sleeve that is slightly longer than the test tube. Simultaneously, position the other tubes under 4000 Lx fluorescent lamps for a reaction period of 10 min, ensuring that the illumination of all tubes is consistent and that the reaction temperature is maintained between 25 and 35°C. At the conclusion of the reaction, terminate it by covering the tubes with a black cloth. Measure the absorbance (A value) of each tube at 560 nm, repeating the measurements three times to calculate SOD activity, using the control tube covered in black as a blank.

SOD activity units are defined as the amount of enzyme required to achieve 50% inhibition of nitroblue tetrazolium (NBT) photochemical reduction, which is considered one unit of

**Table 1. Amount of each solution for colour reaction.**

| reagents (diastase) | dosage/mL | final concentration (colorimetric) |
|---|---|---|
| 0.05 mol/L phosphate buffer | 1.5 | |
| 130 mmol/L Met solution | 0.3 | 13 mmol/L |
| 750 μmol/L NBT solution | 0.3 | 75 μmol/L |
| 100 μmol/L EDTA-Na$_2$ solution | 0.3 | 10 μmol/L |
| 20 μmol/L riboflavin | 0.3 | 2.0 μmol/L |
| digestate | 0.05 | 2 control tubes with buffer solution instead of enzyme solution |
| distilled water | 0.25 | |
| total volume | 3 | |

enzyme activity. SOD activity was calculated using the following formula:

$$\text{SOD activity (u/g)} = \frac{(A_{CK} - A_E)V_T}{0.5A_{CK}WV_S}$$

Where: $A_{CK}$-absorbance of the illuminated control tube;
$A_E$-absorbance of the sample tube;
$V_T$-total volume of sample solution, mL;
$V_S$-volume of sample used in the determination, mL;
W-sample mass, g.

**Determination of peroxidase activity.** POD activity was assessed using the guaiacol method [31]. One gram each of cut and mixed buds, flowers, and young fruits was placed into a mortar. An appropriate volume of phosphate buffer was added, and the mixture was ground into a homogenate. The homogenate was then centrifuged at 4,000 r/min for 15 min. The supernatant was transferred to a 100 mL volumetric flask, and the residue was extracted again with 5 mL of phosphate buffer. This second supernatant was also added to the volumetric flask, and the total volume was adjusted to the calibration mark. For the controls, 3 mL of the reaction mixture and 1 mL of phosphate buffer were added to the first colorimetric cup, while 3 mL of the reaction mixture and 1 mL of the enzyme solution were added to the second color-imetric cup. Add 3 mL of the reaction mixture and 1 mL of the enzyme solution specified above into the No. 2 colorimetric cup to serve as a control. Immediately start the stopwatch to record the time, and measure the absorbance at a wavelength of 470 nm using the spectropho-tometer. Record the absorbance value every minute. Repeat this procedure three times. The results will be calculated using the following formula:

$$\text{POD activity (u/g)} = \frac{\Delta A_{470}V_T}{0.01WV_S t}$$

Where: $A_{470}$-change in absorbance during the reaction time;
W-mass of the sample, g;
$V_T$-total volume of extracted enzyme solution, mL;
$V_S$-volume of enzyme solution taken for determination, mL;
t-reaction time, min.

**Determination of catalase activity.** CAT activity was determined using UV spectropho-tometry [31]. Weigh 0.5 g each of cut and mixed buds, flowers, and young fruits, and place them in a mortar. Add 2 to 3 mL of pre-cooled phosphate buffer (pH 7.8) at 4°C, along with a small amount of quartz sand, and grind the mixture into a homogeneous slurry. Transfer the slurry to a 25 mL volumetric flask, rinsing the mortar several times with buffer to ensure that all material is collected. Combine the buffer washings and adjust the solution to the mark on the flask. Mix thoroughly and place the volumetric flask in a refrigerator at 5°C for 10 min. Afterward, centrifuge the mixture at 4000 r/min for 15 min to obtain the supernatant, which will serve as the crude peroxidase extract. Store the supernatant at 5°C for later use. Prepare three 10 mL test tubes: two will be used for sample determination, and one will serve as a blank (in which the enzyme solution will be boiled to inactivate the enzyme). Add the reagents in the order specified in the Table 2. After preheating to 25°C, 0.3 mL of 0.1 mol/L $H_2O_2$ was added to each tube individually. The timing for each tube commenced immediately after the addition, and each tube was promptly placed into a quartz colorimetric cuvette. The absor-bance was measured at 240 nm, with readings taken at 1-minute intervals for a total duration of 4 min. Enzyme activity was calculated after measuring the absorbance of all three tubes. The results were determined using the following formula:

**Table 2. UV-absorbent samples to be measured solution preparation table.**

| Reagent (enzyme) volume/mL | tube number | | |
|---|---|---|---|
| | S0 | S1 | S2 |
| crude enzyme solution | 0.2 | 0.2 | 0.2 |
| pH = 7.8 phosphate buffer | 1.5 | 1.5 | 1.5 |
| distilled water | 1 | 1 | 1 |

The decrease of 0.1 in $A_{240}$ over 1 minute was defined as 1 unit of enzyme activity (u).

$$\text{CAT activity (u/g)} = \frac{\Delta A_{240}\, V_T}{0.1 V_S t W}$$

Where: $\Delta A_{240} = A_{S0} - \frac{A_{S1}+A_{S2}}{2}$

$A_{S0}$ = absorbance of control tubes to which boil dead enzyme solution was added;

$A_{S1}$, $A_{S2}$-absorbance of sample tubes;

W-sample mass, g;

$V_T$-total volume of crude enzyme extract, mL;

$V_S$-volume of crude enzyme extract for determination, mL;

W-sample mass, g;

Each 0.1 decrease in A240 from 0.1 to 1 min is 1 enzyme activity unit, u;

t-time from addition of hydrogen peroxide to the last reading, min.

**Determination of ascorbic acid content.** AsA content was determined using the bipyridine color development method [31]. First, weigh 0.5 g of clipped and mixed buds, flowers, and young fruits. Add 10 mL of 5% TCA to the samples, then grind and centrifuge at 15,000 g for 10 min. The supernatant is then diluted to a final volume of 25 mL. From this supernatant, 0.2 mL is taken and mixed with 0.2 mL of deionized water, followed by the addition of 0.2 mL of sodium dihydrogen phosphate ($NaH_2PO_4$) solution (pH 7.4) and another 0.2 mL of deionized water. This mixture is thoroughly mixed for 30 seconds. Subsequently, add 0.4 mL of 10% TCA solution, 0.4 mL of phosphoric acid ($H_3PO_4$) solution, 0.4 mL of 2,2-dipyridyl solution, and 0.2 mL of 3% ferric chloride ($FeCl_3$) solution in that order. The optical density (OD) values are measured at 525 nm at 37°C for 60 min. Repeat this procedure three times. The AsA content in the sample is calculated using a standard curve.

**Determination of glutathione content.** GSH content was determined using the 5,5'-dithiobis(2-nitrobenzoic acid) (DTNB) colorimetric method [31]. A total of 0.2 g of plant material was weighed and mixed with 3 mL of pre-cooled 5% trichloroacetic acid solution. The mixture was ground and then adjusted to a final volume of 6 mL. The sample was subjected to centrifugation at 10,000 g for 10 minutes at 4°C, after which the supernatant was collected as the mother liquor and stored at 5°C for later use. From the mother liquor, 0.2 mL was taken and combined with 2.6 mL of 150 mmol/L $NaH_2PO_4$ solution (pH 7.7). The mixture was thoroughly mixed, and then 0.2 mL of DTNB solution was added. The reaction was incubated at 30°C for 5 min, and the absorbance was measured at 412 nm. This procedure was repeated three times, and the GSH content was calculated based on the standard curve equation.

**Determination of the cold resistance gene *VcCBF3*.** Real-time fluorescence quantitative PCR was used for analysis. The primers are listed in Table 3. Total RNA extraction was carried out using the TRIzol kit (Invitrogen) following the kit instructions [32], and the extracted RNA was treated with DNase I. The quality of RNA was assessed using OD assay in combination with agarose gel electrophoresis. The tested and quantified total RNA was reverse

**Table 3. *VcCBF3* gene specific primers for Real-time qPCR analysis.**

| Primer name | Sequence (5' to 3') |
| --- | --- |
| *VcCBF3*-F | AGTGAGGAGGAGGAACAACG |
| *VcCBF3*-R | GACTCACTTTTCAGCGCCAA |
| gapdh-F(nei) | TCTGCCCCAAGTAAGGAT |
| gapdh-R(nei) | TGGAGACAATGTGAAGATCG |

transcribed into cDNA. In an ice bath, 1 μg of total RNA, 1 μl of primer Oligo(dT) (50 uM), 1 μl of dNTP Mix (10 mmol/L), and 1 μl of RNase-free dH$_2$O were added to the tubes to make a total volume of 10 μl. The tubes were mixed well and then incubated at 65˚C for 5 min, followed by a quick ice bath at the end of the incubation. Continue to add 5×Reaction Buffer (4 μl), RNase Inhibitor (40 U/μl) (0.5 μl), MMLV RT (200 U/μl) (1 μl), and RNase-free dH$_2$O to make a total volume of 20 μl. The reaction mixture was heated at 42˚C for 30–60 min, followed by heating at 95˚C for 5 min, and then placed on ice for the subsequent experiments. The reaction mixture was heated at 95˚C for 5 min after being kept on ice for 42–60 min for subsequent experiments.

For each target gene and housekeeping gene, a cDNA template corresponding to the sample was selected for PCR (reaction A): 10 μL of 2×SYBR real-time PCR premixture, 0.4 μL of 10 μM PCR-specific primer F, 0.4 μL of 10 μM PCR-specific primer R, 1 μL of cDNA, and RNase-free dH$_2$O to 20 μL. dH$_2$O to 20 μl. The PCR reaction solution was prepared according to reaction system A and loaded onto a Real-time PCR instrument for PCR reaction. The reaction procedure included pre-denaturation at 95˚C for 5 min, denaturation at 95˚C for 15 seconds, and annealing at 60˚C for 30 seconds. A total of 40 cycles were conducted with three replicates for each sample. Repeat this procedure three times. The data were analyzed using the 2$^{-\Delta\Delta Ct}$ method.

**Data analysis.** A one-way ANOVA and the Waller-Duncan test were employed to analyze the data. The data were imported into IBM SPSS version 26.0 to assess significant differences and were graphically represented using Origin 2022 and Microsoft Excel 2023 software.

## Results and discussion

### Effects of different concentrations of exogenous EBR treatments on osmoregulatory substances during flowering and fruiting in blueberry under low temperature stress

**Soluble sugar content.** As can be seen from Fig 1, under low temperature stress, SS content of blueberry buds, flowers, and young fruits increased to varying degrees compared with the control after spraying exogenous EBR. The overall trend was an initial increase followed by a decrease with the rise in the sprayed EBR concentration. The SS content of buds, flowers, and young fruits reached their peak values at 0.4 mg·L$^{-1}$, which were 3.03%, 7.26%, and 5.52%, respectively. These values were significantly different from the control ($p<0.05$), showing increases of 27.4%, 43.2%, and 142.1%, respectively. The above indicates that spraying exogenous EBR can enhance the accumulation of SS under low temperature stress. This enhancement can improve the osmotic protection capacity of blueberry buds, flowers, and young fruits under low temperature stress, thereby enhancing their cold resistance. In addition, the size of soluble solids content was flower > young fruit > bud.

**Soluble protein content.** As shown in Fig 2, under low temperature stress, SP content of blueberry buds, flowers, and young fruits increased to varying degrees compared to the control after spraying exogenous EBR. They exhibited a trend of initially increasing and then

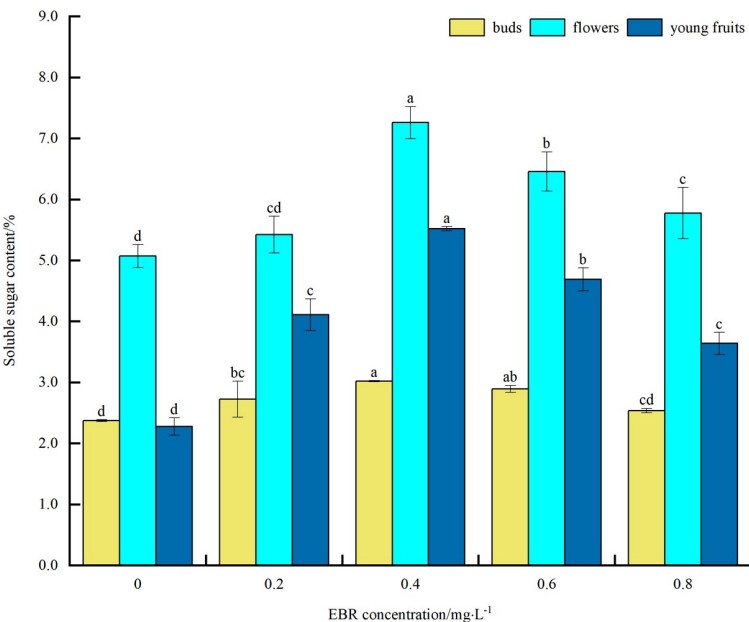

**Fig 1. Effects of exogenous EBR treatment on soluble sugar in blueberry buds, flowers and young fruits.**

decreasing with the rise in the sprayed EBR concentration. The SP content of buds, flowers, and young fruits reached maximum values of 505, 1016, and 449 μg·g$^{-1}$, respectively, at the spraying concentration of 0.4 mg·L$^{-1}$. These values were significantly different from the control ($p<0.05$) and increased by 52.1%, 57.5%, and 89.5%, respectively. The above indicates that spraying exogenous EBR promotes the accumulation of SP under low temperature stress, thereby alleviating the damage of low temperature stress on blueberry buds, flowers, and

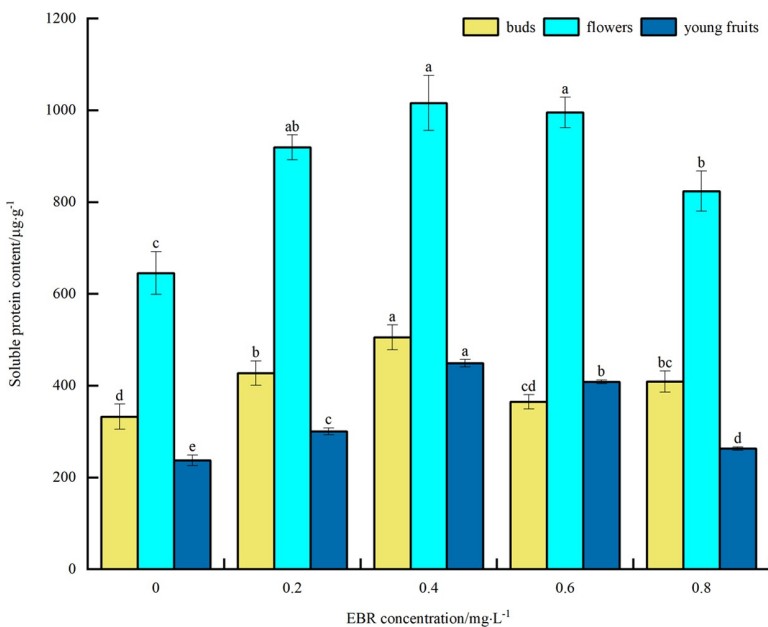

**Fig 2. Effect of exogenous EBR treatment on soluble proteins of blueberry buds, flowers and young fruits.**

young fruits, and improving their cold resistance. In addition, the size of SP content was flower > bud > young fruit.

**Proline content.**   As shown in Fig 3, under low temperature stress, Pro content in blueberry buds, flowers, and young fruits increased to varying degrees compared to the control after applying exogenous EBR. The overall trend was an initial increase followed by a decrease with the rise in the concentration of sprayed EBR. The Pro content of buds, flowers, and young fruits reached the maximum values of 464, 149, and 50 µg·g$^{-1}$ at the spraying concentration of 0.4 mg·L$^{-1}$, respectively. These values were significantly different ($p<0.05$) from the control, showing increases of 78.5%, 204.1%, and 233.3%, respectively. The statement above suggests that spraying exogenous EBR promotes the accumulation of Pro under low-temperature stress. This leads to maintaining cellular osmotic balance in blueberry buds, flowers, and young fruits, thereby enhancing their cold tolerance. In addition, the size of the Pro content varied in the order of bud > flower > young fruit.

## Effects of different concentrations of exogenous EBR treatments on the antioxidant system of blueberry during flowering and fruiting under low temperature stress

**Malondialdehyde content.**   As can be seen in Fig 4, under low temperature stress, MDA content of blueberry buds, flowers, and young fruits decreased to varying degrees compared with the control after spraying exogenous EBR. The overall trend was a decrease followed by an increase with the rise in the sprayed EBR concentration. The MDA content of buds, flowers, and young fruits reached a minimum at 0.4 mg·L$^{-1}$, with 0.011, 0.013, and 0.011 µmol·g$^{-1}$, respectively. These values were significantly different ($p<0.05$) from the control and decreased by 26.7%, 51.9%, and 54.2%, respectively. The above indicates that spraying exogenous EBR would inhibit the accumulation of MDA under low temperature stress, thereby alleviating the damage of low temperature stress on the peroxidation process of the plasma membrane of

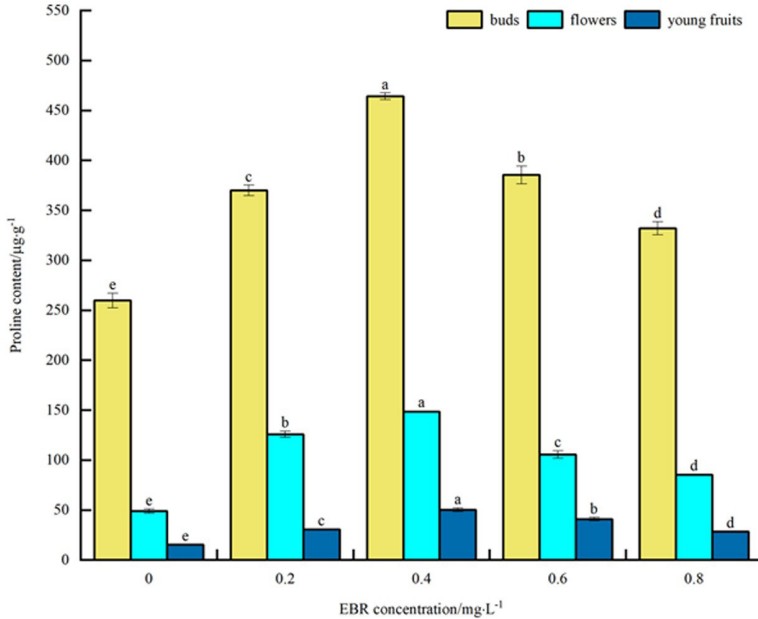

**Fig 3.  Effect of exogenous EBR treatment on proline in blueberry buds, flowers and young fruits.**

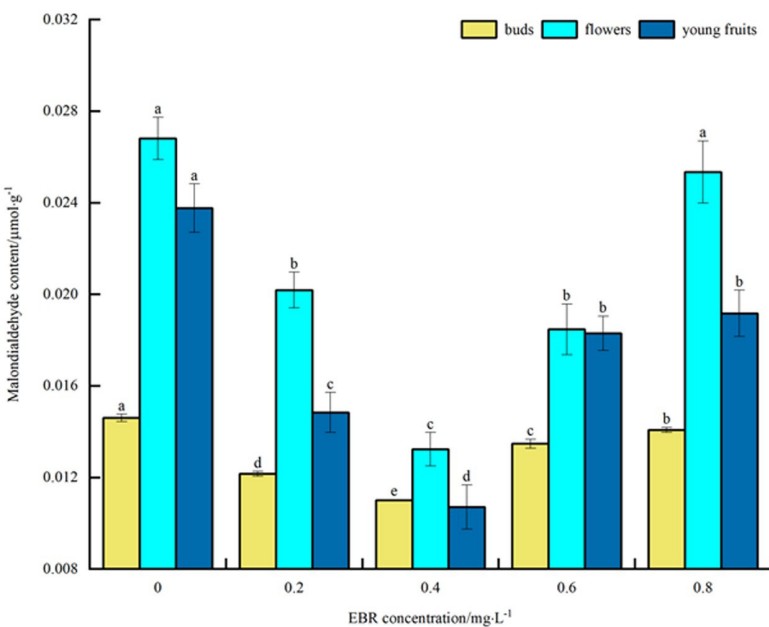

**Fig 4. Effect of exogenous EBR treatment on malondialdehyde in blueberry buds, flowers and young fruits.**

blueberry buds, flowers, and young fruits, and improving their cold resistance. In addition, the MDA content size was ranked as follows: flower > bud > young fruit.

**Rate of superoxide anion production.** As shown in Fig 5, the rate of $O^{2.-}$ production in blueberry buds, flowers, and young fruits decreased to varying degrees compared to the control under low temperature stress after the application of exogenous EBR. The trend overall was a decrease followed by an increase with the rise in the concentration of the sprayed EBR. At 0.4 mg·L$^{-1}$, the $O^{2.-}$ production rates of buds, flowers, and young fruits reached the minimum values of 0.36, 0.47, and 1.13 μmol·min$^{-1}$·g$^{-1}$, respectively. These values were significantly different from the control ($p<0.05$), with decreases of 75.8%, 79.5%, and 48.2%, respectively. The above indicates that spraying exogenous EBR inhibits the accumulation of $O^{2.-}$ under low temperature stress, thereby alleviating the plasma membrane oxidation of blueberry buds, flowers, and young fruits caused by low temperature stress and enhancing their cold resistance. In addition, the magnitude of $O^{2.-}$ production rate was highest in young fruit, followed by flower and bud.

**Superoxide dismutase activity.** As can be seen from Fig 6, under low temperature stress, SOD activity of blueberry buds, flowers, and young fruits increased to varying degrees compared to the control after spraying exogenous EBR. The overall trend was initially upward and then downward with the increase in the concentration of sprayed EBR. The SOD activities of buds, flowers, and young fruits reached their maximum values of 532, 245, and 170 U·g$^{-1}$·min$^{-1}$, respectively, at the spraying concentration of 0.4 mg·L$^{-1}$. These values were significantly different ($p<0.05$) from the control and increased by 36.4%, 27.6%, and 61.9%, respectively. The above indicates that spraying exogenous EBR increased SOD activity in blueberry buds, flowers, and young fruits under low temperature stress, thereby enhancing their cold tolerance. In addition, the magnitude of SOD activity was highest in buds, followed by flowers, and then young fruit.

**Peroxidase activity.** As can be seen from Fig 7, under low temperature stress, POD activity of blueberry buds, flowers, and young fruits increased to varying degrees after spraying

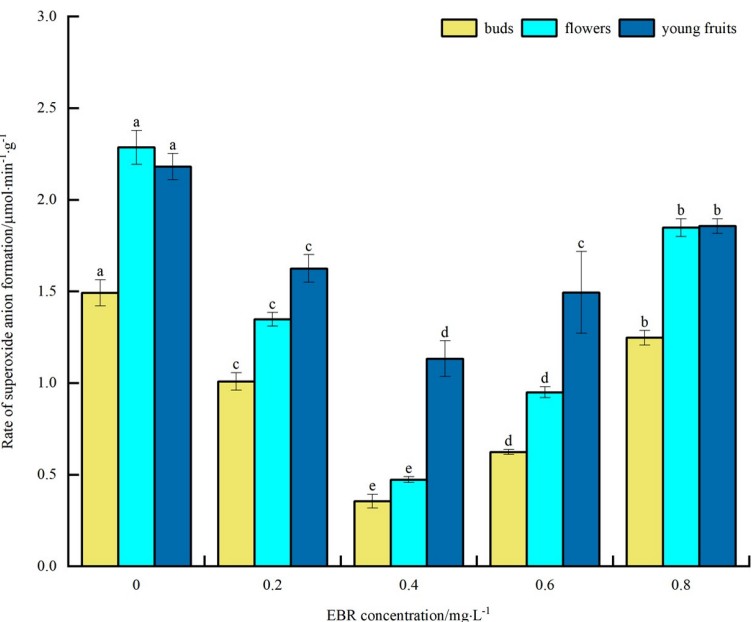

**Fig 5. Effect of exogenous EBR treatment on the rate of superoxide anion production in blueberry buds, flowers and young fruits.**

exogenous EBR compared with the control. The overall trend was initially upward and then downward with the increase in the concentration of the sprayed EBR. The POD activities of buds, flowers, and young fruits reached their maximum values of 97.0, 43.8, and 43.6 U·g$^{-1}$·min$^{-1}$, respectively, at a spraying concentration of 0.4 mg·L$^{-1}$. These values were significantly different ($p<0.05$) from the control, showing an increase of 1963.8%, 45.5%, and 285.8%, respectively. The above indicates that spraying exogenous EBR increased POD activity in blueberry buds, flowers,

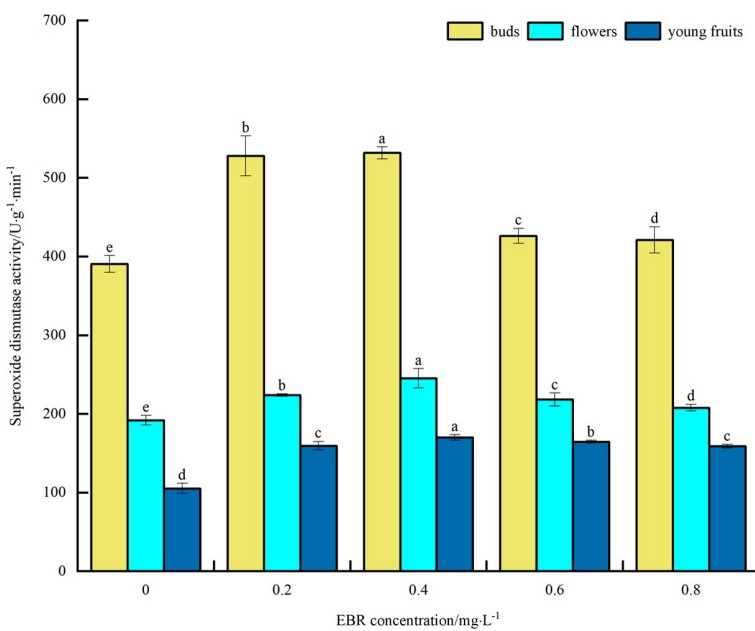

**Fig 6. Effect of exogenous EBR treatment on superoxide dismutase in blueberry buds, flowers and young fruits.**

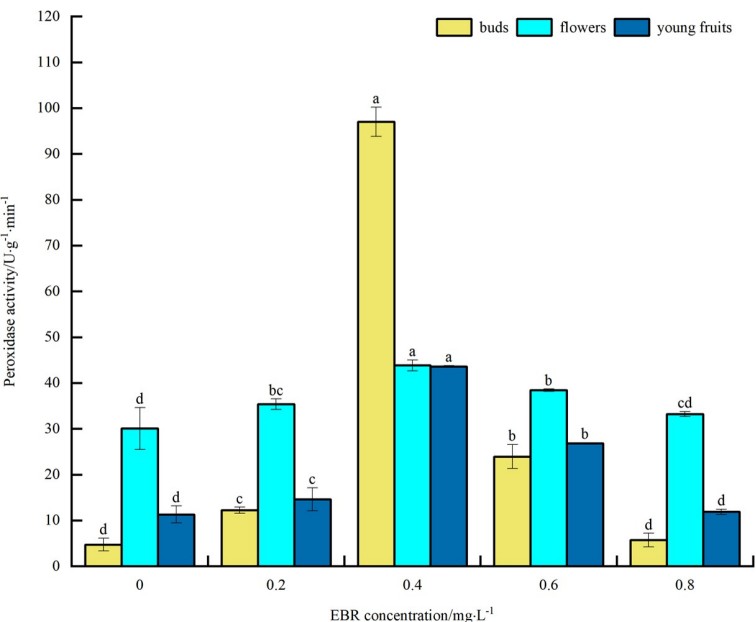

**Fig 7. Effect of exogenous EBR treatment on peroxidase in blueberry buds, flowers and young fruits.**

and young fruits under low temperature stress, thereby enhancing their cold tolerance. In addition, the magnitude of POD activity was highest in buds, followed by flowers, and then young fruit.

**Catalase activity.** As can be seen from Fig 8, under low-temperature stress, CAT activity of blueberry buds, flowers, and young fruits increased to varying degrees after spraying exogenous EBR compared with the control. The overall trend was initially upward and then

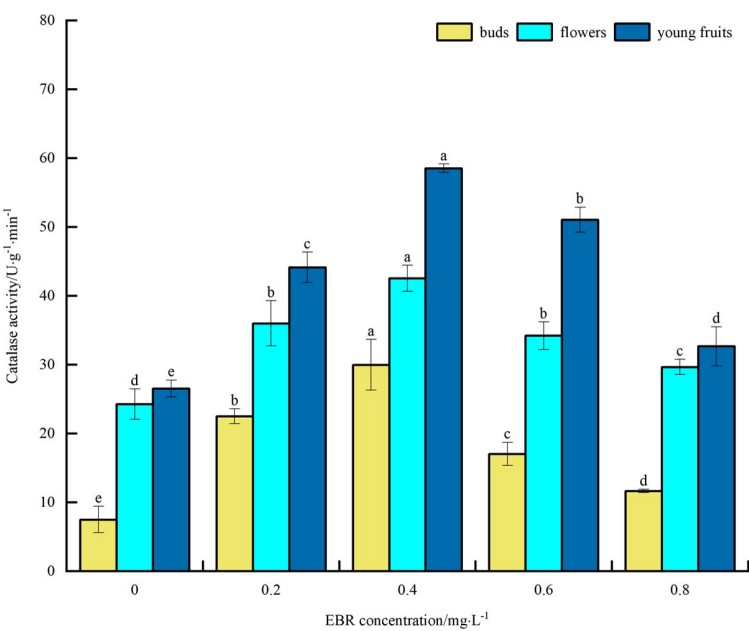

**Fig 8. Effects of exogenous EBR treatment on catalase in blueberry buds, flowers and young fruits.**

downward with the increase in the sprayed EBR concentration. The CAT activities of buds, flowers, and young fruits reached their maximum values of 30.0, 42.5, and 58.5 U·g$^{-1}$·min$^{-1}$, respectively, at the spraying concentration of 0.4 mg·L$^{-1}$. These values were significantly different ($p < 0.05$) from the control, showing increases of 300.0%, 74.9%, and 120.8%, respectively. The above indicates that spraying exogenous EBR increased CAT activity in blueberry buds, flowers, and young fruits under low temperature stress, thereby enhancing their cold tolerance. In addition, the level of CAT activity was highest in young fruit, followed by flower and bud.

**Ascorbic acid content.** As can be seen from Fig 9, under low temperature stress, AsA content of blueberry buds, flowers, and young fruits increased to varying degrees compared to the control after spraying exogenous EBR. The overall trend showed an initial increase followed by a decrease with the rise in the concentration of sprayed EBR. The AsA contents of flower buds, flowers, and young fruits reached maximum values of 45.1, 48.8, and 48.8 μg·g$^{-1}$ at the spraying concentration of 0.4 mg·L$^{-1}$, which were significantly different ($p < 0.05$) from the control. They increased by 48.4%, 43.1%, and 17.6%, respectively. The above indicates that spraying exogenous EBR increased AsA content in blueberry buds, flowers, and young fruits under low temperature stress, thereby enhancing their cold tolerance. In addition, the size of AsA content was highest in flowers, followed by young fruit, and then buds.

**Glutathione content.** As can be seen from Fig 10, under low temperature stress, GSH content of blueberry buds, flowers, and young fruits increased to varying degrees compared with the control after spraying exogenous EBR. The overall trend was an initial increase followed by a decrease with the rise in the sprayed EBR concentration. The GSH contents of buds, flowers, and young fruits reached their maximum values at 0.4 mg·L$^{-1}$, measuring 90.8 μg·g$^{-1}$, 179.8 μg·g$^{-1}$, and 638.8 μg·g$^{-1}$, respectively. These values were significantly different from the control ($p < 0.05$), showing increases of 57.9%, 83.8%, and 88.8%, respectively. The above indicates that spraying exogenous EBR increased the GSH content in blueberry buds, flowers, and young fruits under low temperature stress, thereby enhancing

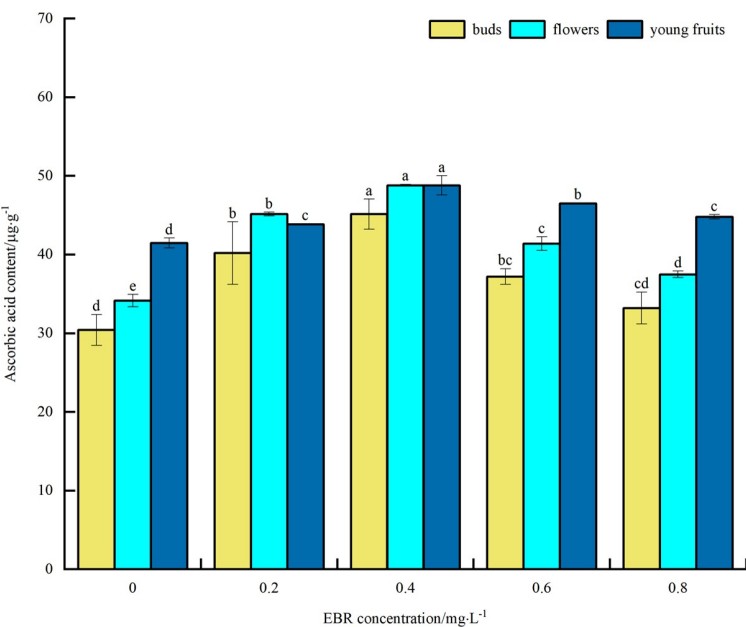

**Fig 9. Effect of exogenous EBR treatment on ascorbic acid content of blueberry buds, flowers and young fruits.**

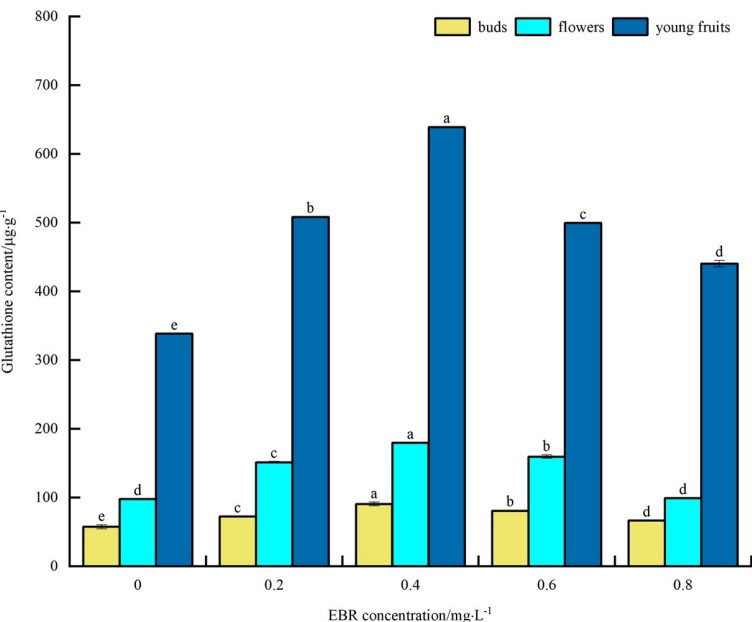

**Fig 10. Effect of exogenous EBR treatment on glutathione content of blueberry buds, flowers and young fruits.**

their cold resistance. In addition, the GSH content was highest in young fruit, followed by flowers, and then buds.

## Effects of different concentrations of exogenous EBR treatments on the expression of the cold-resistant gene *VcCBF3* during the flowering and fruiting stages of blueberry under low temperature stress

As can be seen from Fig 11, under low temperature stress, the relative expression of the *VcCBF3* gene in blueberry buds, flowers, and young fruits increased to varying degrees compared with the control after spraying exogenous EBR. It showed an overall trend of increasing and then decreasing with the increase in the concentration of sprayed EBR. The relative expression of the *VcCBF3* gene in buds, flowers, and young fruits reached its maximum at 0.4 mg·L$^{-1}$, with values of 7.4, 3.1, and 4.7, respectively. These values showed significant differences ($p<0.05$) compared to the control and increased by 362.5%, 106.7%, and 365.0%, respectively. The above indicates that spraying exogenous EBR increased the relative expression of the *VcCBF3* gene in blueberry buds, flowers, and young fruits under low temperature stress, thereby enhancing their cold resistance. In addition, the relative expression size of the *VcCBF3* gene was highest in buds, followed by young fruit, and then flowers.

## Comprehensive evaluation of cold tolerance using the affiliation function method

As shown in Table 4, the overall cold resistance under various concentrations of exogenous EBR treatments was ranked as follows: flowers > buds > young fruits. Buds, flowers, and young fruits sprayed with EBR at a concentration of 0.4 mg·L$^{-1}$ exhibited higher cold resistance compared to other concentrations. The average affiliation degrees were 0.677, 0.673, and 0.617 for buds, flowers, and young fruits, respectively. These values corresponded to cold-resistant classes II, II, and II, respectively.

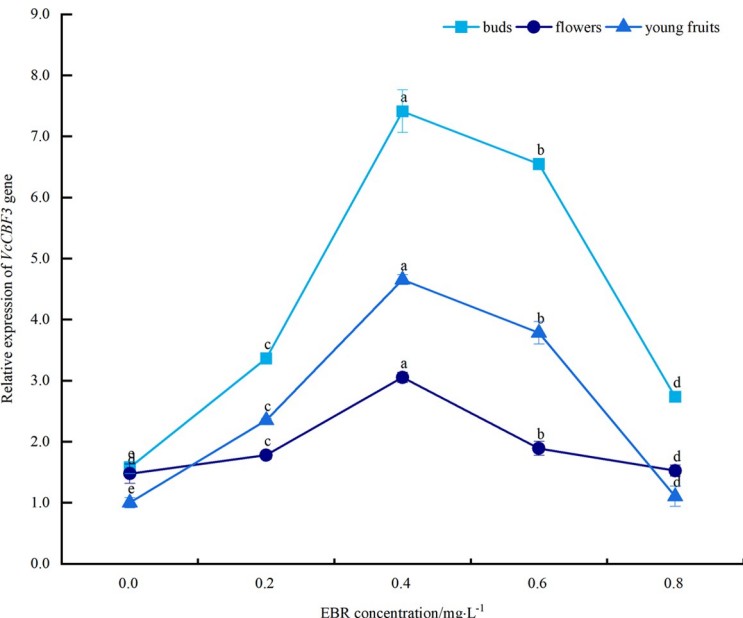

**Fig 11. Effect of exogenous EBR treatment on the relative expression of *VcCBF3* gene in blueberry buds, flowers and young fruits.**

## Correlation analysis of the exogenous EBR treatment on nine physiological indicators and *VcCBF3* gene expression in blueberries under low temperature stress

As depicted in Fig 12, the correlation between exogenous EBR treatments at various concentrations on nine physiological indexes and *VcCBF3* gene expression in blueberries under low temperature stress was analyzed. It was observed that SS content exhibited a highly significant

**Table 4. Comprehensive evaluation of cold resistance at flowering and fruiting stage of blueberry treated with exogenous EBR under artificial low temperature stress.**

| Place | Treatment concentration | Membership degree of each cold resistance index | | | | | | | | | | Average affiliation | Types of Cold resistance |
|---|---|---|---|---|---|---|---|---|---|---|---|---|---|
| | | SS | Sp | Pro | MDA | $O^{2-}$ | SOD | POD | CAT | AsA | GSH | | |
| **Buds** | 0 | 0.019 | 0.122 | 0.545 | 0.758 | 0.411 | 0.669 | 0.000 | 0.000 | 0.000 | 0.000 | 0.252 | S |
| | 0.2 | 0.090 | 0.244 | 0.790 | 0.910 | 0.662 | 0.991 | 0.081 | 0.294 | 0.531 | 0.026 | 0.462 | MR |
| | 0.4 | 0.149 | 0.344 | 1.000 | 0.981 | 1.000 | 1.000 | 1.000 | 0.441 | 0.800 | 0.057 | 0.677 | R |
| | 0.6 | 0.124 | 0.164 | 0.825 | 0.828 | 0.861 | 0.753 | 0.208 | 0.187 | 0.368 | 0.040 | 0.436 | MR |
| | 0.8 | 0.053 | 0.220 | 0.706 | 0.791 | 0.538 | 0.741 | 0.010 | 0.081 | 0.152 | 0.016 | 0.331 | LR |
| **Flowers** | 0 | 0.561 | 0.524 | 0.075 | 0.000 | 0.000 | 0.203 | 0.275 | 0.329 | 0.203 | 0.069 | 0.224 | S |
| | 0.2 | 0.631 | 0.876 | 0.246 | 0.411 | 0.486 | 0.278 | 0.332 | 0.558 | 0.800 | 0.161 | 0.478 | MR |
| | 0.4 | 1.000 | 1.000 | 0.297 | 0.843 | 0.939 | 0.328 | 0.424 | 0.686 | 1.000 | 0.210 | 0.673 | R |
| | 0.6 | 0.839 | 0.973 | 0.201 | 0.518 | 0.692 | 0.265 | 0.365 | 0.523 | 0.596 | 0.176 | 0.515 | MR |
| | 0.8 | 0.702 | 0.753 | 0.156 | 0.091 | 0.227 | 0.241 | 0.309 | 0.434 | 0.384 | 0.072 | 0.337 | LR |
| **Young fruits** | 0 | 0.000 | 0.000 | 0.000 | 0.189 | 0.054 | 0.000 | 0.071 | 0.373 | 0.601 | 0.483 | 0.177 | S |
| | 0.2 | 0.368 | 0.081 | 0.034 | 0.744 | 0.342 | 0.128 | 0.107 | 0.718 | 0.731 | 0.776 | 0.403 | MR |
| | 0.4 | 0.651 | 0.272 | 0.078 | 1.000 | 0.597 | 0.152 | 0.421 | 1.000 | 0.998 | 1.000 | 0.617 | R |
| | 0.6 | 0.484 | 0.220 | 0.057 | 0.529 | 0.410 | 0.139 | 0.239 | 0.854 | 0.875 | 0.760 | 0.457 | MR |
| | 0.8 | 0.274 | 0.033 | 0.029 | 0.475 | 0.222 | 0.126 | 0.077 | 0.493 | 0.782 | 0.659 | 0.317 | LR |

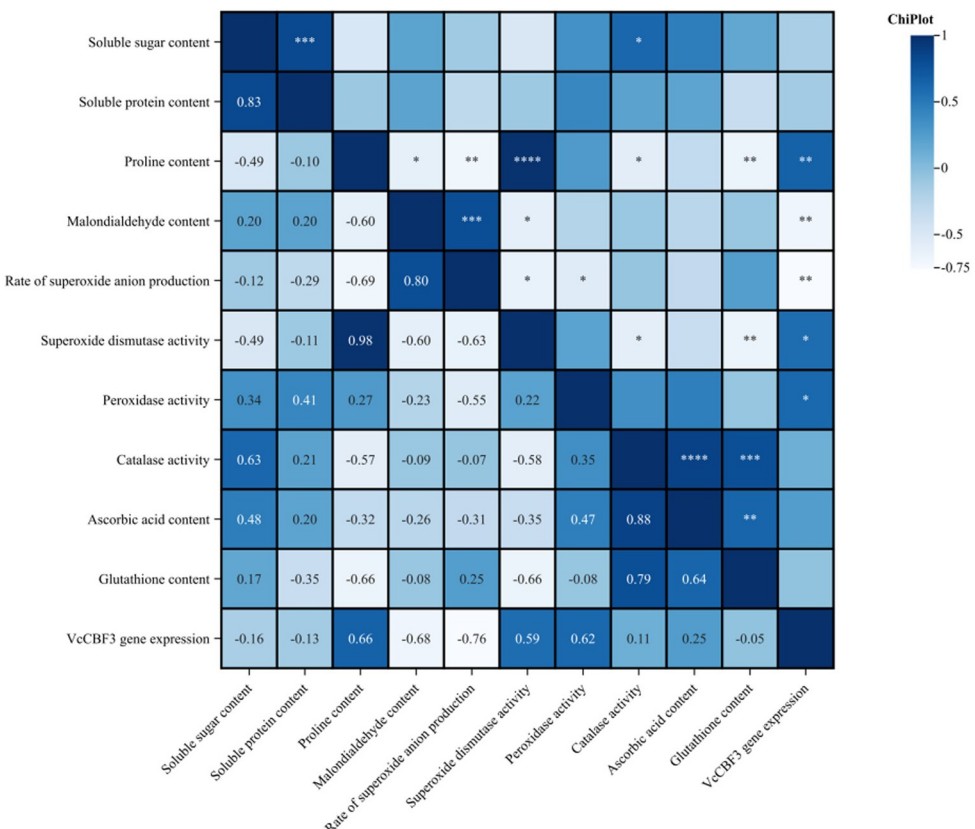

**Fig 12. Correlation analysis of exogenous EBR treatment on 9 physiological indexes and *VcCBF3* gene expression in buds, flowers and young fruits of blueberry.**

positive correlation with SP content and CAT activity. Additionally, Pro content displayed a highly significant negative correlation with MDA content and CAT activity, while showing a highly significant positive correlation with the rate of $O^{2.-}$ production, GSH content, SOD activity, and *VcCBF3* gene expression. Protein content showed a significant negative correlation with MDA content, CAT activity, and the rate of $O^{2.-}$ production, a significant negative correlation with GSH content, a significant positive correlation with SOD activity, and *VcCBF3* gene expression. MDA content showed a significant negative correlation with the rate of $O^{2.-}$ production, a significant negative correlation with SOD activity, and a significant negative correlation with *VcCBF3* gene expression. The rate of $O^{2.-}$ production showed a significant negative correlation with SOD activity. The rate of $O^{2.-}$ production showed a significant negative correlation with SOD activity, POD activity, and *VcCBF3* gene expression. SOD activity showed a significant negative correlation with CAT activity, a significant negative correlation with GSH content, and a significant positive correlation with *VcCBF3* gene expression. POD activity showed a significant positive correlation with *VcCBF3* gene expression. CAT activity showed a significant positive correlation with AsA content and GSH content. AsA content showed a significant positive correlation with GSH content. The POD activity showed a significant and positive correlation with *VcCBF3* gene expression. Except for the above-mentioned indicators, there was no significant relationship among the other variables, suggesting a lack of correlation between them.

## Discussion

Under low temperature stress, plants accumulate organic or inorganic substances to enhance resistance against the damage caused by low temperatures. This process is known as osmoregulation in plants. SS, SP, and Pro are crucial osmoregulators in plants. They play a vital role in protecting cell membrane permeability, maintaining cell membrane stability, and reducing osmotic potential. In this experiment, the levels of osmoregulatory substances in blueberry buds, flowers, and young fruits under low temperature stress exhibited an increasing trend to resist more severe low temperature damage. However, this trend started to decrease once a certain threshold was reached. These findings align with the results of a study by Ji HY on lingonberry 'Northland' [4]. It was shown that spraying EBR could improve the osmoregulation ability of plants by regulating osmotic substances, which was consistent with the results of Li Y in winter radish [33]. In this experiment, the levels of SP, SS, and Pro peaked at a spraying concentration of 0.4 mg·L$^{-1}$, showing the most significant increase. This finding aligns with Dou LL's study, which suggested that EBR could enhance the cold hardiness of cotton seedlings [34]. However, Dou LL proposed that the optimal concentration for improving cold hardiness in cotton was 0.2 mg/L, possibly due to interspecies differences.

When plants are subjected to low temperature stress, a large amount of reactive oxygen species, such as $O^{2 \cdot -}$, accumulates in the cells. This accumulation leads to an imbalance in reactive oxygen species metabolism, resulting in oxidative damage to the cell membrane system. When exposed to stressful conditions, the production of free radicals increases while the scavenging rate decreases due to disruptions in their physiological functions. This leads to the accumulation of free radicals, causing damage to the cell membrane. Consequently, there is an elevation in membrane lipid peroxidation, with MDA being the resultant product [35]. To mitigate the damage, the activity of the antioxidant defense system increases in plants. The antioxidant system includes both antioxidant enzyme systems and non-enzymatic antioxidant systems. The antioxidant enzymes SOD, POD, CAT, and the AsA-GSH cycle are effective in scavenging excess accumulated reactive oxygen species. In this study, the activities of SOD, POD, and CAT were increased in blueberry buds, flowers, and young fruits at low temperatures. Spraying EBR was found to enhance the activities of these enzymes and reduce the content of oxygen free radicals. This suggests that EBR induced an increase in the activities of antioxidant enzymes, which could effectively eliminate the accumulated reactive oxygen species, decrease membrane lipid peroxidation, and enhance the cold hardiness of blueberries. This is in agreement with the findings of Prabhu D, who pointed out that exogenous EBR could improve the scavenging capacity of reactive oxygen species, thereby enhancing the cold tolerance of Phragmites australis [36]. The decrease in MDA content and the rate of $O^{2 \cdot -}$ production after EBR spraying was consistent with the findings of Wang CQ et al. in dragon fruit seedlings [37]. They highlighted that exogenous EBR treatment reduced the rate of $O^{2 \cdot -}$ production and the content of MDA, a byproduct of membrane lipid peroxidation, effectively enhancing the cold tolerance of dragon fruit seedlings. In this study, exogenous EBR up-regulated the AsA-GSH cycling system, which is consistent with the results of Chen ZY et al. in grape seedlings [38]. They noted that EBR accelerated the AsA-GSH cycling and enhanced ROS scavenging by promoting the relevant enzymes in the cycle.

The primary pathway of the cold-resistance mechanism in plants induced by the *CBF (CRT/DRE-binding factor)* gene is as follows: when plants are exposed to low temperatures, the *ICE* (Inducer of *CBF* Expression) transcription factor is activated. This factor binds to the *ICE* box situated in the promoter region upstream of the *CBF* gene, thereby triggering the expression of the *CBF* gene. The product of the *CBF* gene binds to the CRT/DRE (C-repeat binding factor/dehydration-responsive element binding protein) element in the promoter of a series of

downstream *COR* genes, leading to the expression of a set of cold-resistant genes. thereby improving the cold resistance of plants [39]. In the present study, the expression of the *VcCBF3* gene was up-regulated after the external application of EBR. This finding is consistent with the results of Ma JH in maize, who suggested that EBR might boost the antioxidant capacity of maize seed embryos. Additionally, it could regulate the expression of low-temperature-responsive genes through the NO pathway, thereby intervening in the enhancement of low temperature stress tolerance [40]. External application of EBR enhanced cold tolerance under low temperature stress in plants. This finding is consistent with the research by Jiang YS on cucumber, which highlighted that CSN enhances cold tolerance by activating EBR signaling [41]. This activation contributes to the gene expression of ICE-CBF-COR. Moreover, the combination of CSN and EBR contributes to cold tolerance and the recovery of CS-damaged seedlings in cucumber.

The correlation function method is a comprehensive evaluation technique based on fuzzy mathematical theory, widely utilized in assessing the cold tolerance of plants. Correlation analysis is a statistical method that describes the degree and nature of the linear relationship between two variables [42]. By analyzing the correlation between 9 physiological indexes and *VcCBF3* gene expression in blueberries under low temperature stress with different concentrations of exogenous EBR treatments, it was found that 9 pairs of indexes reached a significant level, and 12 pairs of indexes reached a highly significant level. This suggests that the indexes are independent of each other and influence each other. In this experiment, buds, flowers, and young fruits sprayed with a concentration of 0.4 mg·L$^{-1}$ had the highest cold resistance, being resistant, while the control group sprayed with clear water exhibited the weakest cold resistance, being non-resistant. The results of the comprehensive evaluation and analysis indicated that externally applied EBR could effectively improve the cold resistance of blueberries during the flowering and fruiting stages. This finding aligns with the results of a similar study on 'Merlot' grapes by Guo XL [18], which highlighted that applying EBR treatment at a concentration of 0.4 mg/L mitigated the damage caused by low temperatures to the seedlings and enhanced their cold resistance. Consistent with the findings of Zhao FF on Merlot grape branches, it was highlighted that the most favorable outcome was observed with the 0.4 mg·L$^{-1}$ treatment under various low temperature stresses [43]. This suggests that the 0.4 mg·L$^{-1}$ treatment is most effective in enhancing cold resistance during the concentration increase period.

## Conclusions

External application of exogenous EBR can effectively alleviate the damage to blueberry growth under low temperature stress. It promotes an increase in osmoregulatory substances content, reduces the damage caused by the plasma membrane peroxidation process, enhances the antioxidant system, and up-regulates the relative expression of the cold-resistant gene, *VcCBF3* gene, thereby enhancing the cold resistance of blueberries. There is a noticeable concentration effect, There was a significant concentration effect observed, within the tested concentrations of EBR—0, 0.2, 0.4, 0.6, and 0.8 mg·L$^{-1}$—the topical application of 0.4 mg·L$^{-1}$ EBR was found to be the most effective in alleviating low-temperature stress injuries in blueberries. Overall, flowers have better resistance to low temperatures compared to buds and young fruits.

## Supporting information

**S1 File. Trizol_reagent.**
(PDF)

**S1 Data.**
(XLSX)

**S2 Data.**
(XLSX)

**S3 Data.**
(XLSX)

**S4 Data.**
(XLSX)

**S5 Data.**
(XLSX)

**S6 Data.**
(XLSX)

**S7 Data.**
(XLSX)

**S8 Data.**
(XLSX)

**S9 Data.**
(XLSX)

**S10 Data.**
(XLSX)

**S11 Data.**
(XLSX)

**S12 Data.**
(XLSX)

## Acknowledgments

Thanks for the PLOS ONE journal for recognizing my paper, and to the academic editor, Dr. Mojtaba Kordrostami, for his prompt and continuous communication regarding the issues raised. I also extend my thanks to the two reviewers for their thorough evaluation of my paper and their valuable feedback.

## Author Contributions

**Data curation:** Jiahe Fan.

**Funding acquisition:** Xuedong Tang.

**Investigation:** Jiaxin Cai, Xue Gao.

**Validation:** Ruiyang Tan.

**Visualization:** Jiahe Fan.

**Writing – original draft:** Jiahe Fan.

**Writing – review & editing:** Jiahe Fan.

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
