## [Decision Letter · Decision Letter 0]

10 Sep 2024

PONE-D-24-22975Effects of exogenous EBR on the physiology of cold resistance and the expression of the CBF3 gene in blueberries during low-temperature stressPLOS ONE

Dear Dr. fan,

Thank you for submitting your manuscript to PLOS ONE. After careful consideration, we feel that it has merit but does not fully meet PLOS ONE’s publication criteria as it currently stands. Therefore, we invite you to submit a revised version of the manuscript that addresses the points raised during the review process.

We look forward to receiving your revised manuscript.

Kind regards,

Mojtaba Kordrostami, Ph.D.

Academic Editor

PLOS ONE

 Please confirm at this time whether or not your submission contains all raw data required to replicate the results of your study. Authors must share the “minimal data set” for their submission. PLOS defines the minimal data set to consist of the data required to replicate all study findings reported in the article, as well as related metadata and methods (https://journals.plos.org/plosone/s/data-availability#loc-minimal-data-set-definition). For example, authors should submit the following data: - The values behind the means, standard deviations and other measures reported; - The values used to build graphs; - The points extracted from images for analysis. Authors do not need to submit their entire data set if only a portion of the data was used in the reported study. If your submission does not contain these data, please either upload them as Supporting Information files or deposit them to a stable, public repository and provide us with the relevant URLs, DOIs, or accession numbers. For a list of recommended repositories, please see https://journals.plos.org/plosone/s/recommended-repositories. If there are ethical or legal restrictions on sharing a de-identified data set, please explain them in detail (e.g., data contain potentially sensitive information, data are owned by a third-party organization, etc.) and who has imposed them (e.g., an ethics committee). Please also provide contact information for a data access committee, ethics committee, or other institutional body to which data requests may be sent. If data are owned by a third party, please indicate how others may request data access.

Reviewers' comments:

Reviewer's Responses to Questions

**Comments to the Author**

1. Is the manuscript technically sound, and do the data support the conclusions?

Reviewer #1: No

Reviewer #2: Yes

2. Has the statistical analysis been performed appropriately and rigorously? 

Reviewer #1: No

Reviewer #2: N/A

3. Have the authors made all data underlying the findings in their manuscript fully available?

Reviewer #1: Yes

Reviewer #2: Yes

4. Is the manuscript presented in an intelligible fashion and written in standard English?

Reviewer #1: Yes

Reviewer #2: Yes

5. Review Comments to the Author

Reviewer #1: 1. What is EBR, authors should properly introduce it in the manuscript before giving it an abbreviation.

2. Cold storage conditions and interval periods are missing in the abstract. please include.

3. Results in the abstract should be compared among the EBR concentrations and conclude with the suitable concentration as recommended in the abstract.

4. Instead of CBF3, the author should replace this cold-resistance gene.

5. The introduction is poorly presented, it lacks reasoning and a hypothetical approach that convinces the tested objective of the work, I recommend a complete rewrite with elaborated details.

6. Section 2.3.1 should be elaborated with detailed methods.

7. Section 2.4 - provides no details! poorly presented.

Reviewer #2: 1. In the introduction section, the authors should tell audiences more information. For example, how does blueberry deal with cold stress? how does EBR affect cold resistance in plant? Why can EBR be applied to blueberry? Additionally, there is a sentence in the first paragraph of introduction part is “The shrub is shallow-rooted and has weak cold resistance”. “Weak cold resistance” should be carefully used to blueberry.

2. In the Material and Methods section 2.2, it was mentioned “The treatment involved a temperature of 2°C, a cooling rate of 4°C/h, an 8-hour maintenance period after reaching the desired temperature, followed by a gradual warming at the same rate.” What is the reason that the treatment temperature is 2°C and the treatment duration is 8-hour? Please explain it in the manuscript. The last two sentences in 2,3.2 should be rewritten. how many replicates should be provided. Based on my knowledge, TRIzol kit (Invitrogen) can not work well for blueberry RNA isolation, please double check it. Additionally, no reference was listed for physiological analysis, which should be cited accordingly.

6. PLOS authors have the option to publish the peer review history of their article (what does this mean?). If published, this will include your full peer review and any attached files.

Reviewer #1: No

Reviewer #2: No

---

## [Author Response · Author response to Decision Letter 0]

11 Oct 2024

Dear Reviewer 1:

We are very grateful to Reviewer for reviewing the paper so carefully. We have tried our best to improve the manuscript and have modified. The revised paragraphs are labeled in yellow. Responds to the Reviewer’s comments were as follow:

Comments 1:

What is EBR, authors should properly introduce it in the manuscript before giving it an abbreviation.

Response 1:

Thank you for pointing this out. We agree with this comment.

We revise the sentence to become “In recent years, exogenous substances have attracted considerable attention for their role in enhancing plant cold resistance. 2,4-Epibrassinolide (EBR) is a novel class of highly effective, broad-spectrum, and environmentally friendly plant growth regulators. It has the capacity to regulate plant growth and development, improve resistance to low temperatures, and is recognized as the sixth major plant hormone on the global stage”

Modification on the page 3, line 98-103.

Comments 2:

Cold storage conditions and interval periods are missing in the abstract. please include.

Response 2:

Thank you for pointing this out. We agree with this comment.

Therefore, we have been added cold storage conditions and interval periods to the abstract. We revise the sentence to become “The northern highbush blueberry variety ‘Duke’ was used as the test material, and different concentrations of 2,4-Epibrassinolide (EBR) (0, 0.2, 0.4, 0.6, and 0.8 mg·L-1) were applied during the bud expansion stage, with a second application administered at one-day intervals following the first. Samples were collected at the bud, flower, and fruit stages and subsequently treated with artificial low temperatures (2°C) after sampling.”

Modification on the page 2, line 44-49.

Comments 3: 

Results in the abstract should be compared among the EBR concentrations and conclude with the suitable concentration as recommended in the abstract.

Response 3:

Thank you for pointing this out. We agree with this comment.

We revise the sentence to become “Appropriate concentrations of exogenous EBR can effectively mitigate freezing damage in blueberries caused by low temperatures. A comprehensive evaluation and correlation analysis of each cold tolerance index and the expression of the VcCBF3 gene revealed that a treatment concentration of 0.4 mg·L-1 had the most significant mitigating effect among the sprayed EBR concentrations of 0, 0.2, 0.4, 0.6, and 0.8 mg·L-1.” and “There was a significant concentration effect observed, within the tested concentrations of EBR—0, 0.2, 0.4, 0.6, and 0.8 mg·L-1—the topical application of 0.4 mg·L-1 EBR was found to be the most effective in alleviating low-temperature stress injuries in blueberries.”

Modification on the page 2, line 62-67 and page 22, line 835-838.

Comments 4: 

Instead of CBF3, the author should replace this cold-resistance gene.

Response 4: 

We are very sorry for our incorrect writing and it has been rectified.

We have replaced the CBF3 gene with the VcCBF3 gene throughout the text.

Comments 5:

The introduction is poorly presented, it lacks reasoning and a hypothetical approach that convinces the tested objective of the work, I recommend a complete rewrite with elaborated details.

Response 5:

Thank you for pointing this out. We agree with this comment.

We revise the introduction to become “Blueberry, which belong to the rhododendron family (Ericaceae) and are classified under the genus Vaccinium, are perennial, shallow-rooted shrubs. They produce nutritious fruits that are both sweet and tart, accompanied by a delightful aroma. Changchun, located in northeastern China within an alpine region, experiences extended periods of low temperatures. In spring, as fruit trees begin to bud and flower, the flower organs exhibit a limited ability to withstand external damage. Temperature fluctuations, characterized by warming followed by cooling, can lead to occur ‘inverted spring cold’ phenomenon, resulting in the drying and wilting of flower buds, browning and dropping of flower organs, and even deformation of young fruits. Such conditions significantly hinder the development of the blueberry industry in the Changchun area. In a comparative study examining the cold resistance of various lingonberry varieties, Ren B concluded that the soluble sugar (SS) content increased as the temperature decreased. The soluble protein (SP) content exhibited an initial increase, followed by a subsequent decrease. Although the free proline (Pro) content increased, it gradually declined with the intensification of stress and the extension of the treatment duration. Furthermore, the malondialdehyde (MDA) content initially rose before slowly declining. Wei X et al. discovered that the MDA content in various blueberry branch varieties gradually increased as the temperature decreased, following an 'S'-shaped curve. In contrast, the activities of peroxidase (POD), superoxide dismutase (SOD), and catalase (CAT) exhibited a unimodal trend, initially increasing before subsequently decreasing. Liu BH et al. observed that the ascorbate-glutathione (AsA-GSH) system demonstrated a pattern of increase followed by a decrease as the temperature declined from the pre-dormant stage to the post-dormant stage.

In recent years, exogenous substances have attracted considerable attention for their role in enhancing plant cold resistance. 2,4-Epibrassinolide (EBR) is a novel class of highly effective, broad-spectrum, and environmentally friendly plant growth regulators. It has the capacity to regulate plant growth and development, improve resistance to low temperatures, and is recognized as the sixth major plant hormone on the global stage. EBR enhances the cold tolerance of plants by reducing the extent of membrane lipid peroxidation. It achieves this by increasing antioxidant capacity, lowering MDA levels, and decreasing the degree of lipid peroxidation in cellular membranes. Consequently, EBR improves the water-holding capacity of cells and tissues, thereby mitigating damage caused by low temperatures. Additionally, EBR modulates plant metabolism by inhibiting the generation of excessive free radicals and promoting the production of free radical scavengers, which stabilizes the structure and function of cellular membranes. EBR treatment also activates the antioxidant enzyme system, enhancing the activity of key antioxidant enzymes such as SOD, POD, and CAT. Furthermore, it induces the accumulation of non-enzymatic antioxidants, including glutathione and proline, which effectively scavenge reactive oxygen species (ROS). It has been demonstrated that EBR can enhance the cold resistance of various plants, including tea trees, peanuts, bananas, loquats, and others, playing a crucial role in mitigating low-temperature stress. Guo XL et al. reported that EBR improved the cold resistance of grape leaves. Similarly, Yi CX et al. found that a 0.5 μM EBR treatment significantly increased the cold resistance of lychee. Additionally, Hu SQ et al. discovered that post-harvest EBR treatment enhanced the cold resistance of peach fruits. Currently, there are no reports on the effects of EBR on the cold resistance of blueberries under low temperature stress in China.

When plants are exposed to low temperature stress, they first detect the signals associated with reduced temperatures and subsequently activate transcription factors through a series of signal transduction pathways. These transcription factors bind to cis-acting elements, resulting in the expression of various low-temperature-responsive genes that produce numerous functional proteins, thereby enhancing the plants' cold tolerance. The transduction pathway for low temperature signaling in plants has been extensively studied, with one of the most prominent pathways being the CBF- dependent pathway (ICE-CBF-COR). CBF transcription factors are part of the AP2/EREBP family, which plays a crucial role in the CBF dependent pathway and is conserved across a wide range of higher plants. Current research indicates that CBFs are regulated by ICE1, ICE2, and the calmodulin-binding transcriptional activator CAMTA3. Notably, the transcription factor family recognizes cold-responsive and dehydration-responsive elements (CRT/DRE) in the COR promoter region, which is why CBF transcription factors are also referred to as dehydration-responsive element-binding factors (DREB). In Arabidopsis thaliana, the CBF gene, which is part of the DREB subfamily, comprises three copies: CBF1, CBF2, and CBF3. All three copies can be regulated by the upstream transcription factor ICE. The activation of CBF results in the upregulation of its downstream COR genes, thereby enhancing the cold resistance of the plants. CBF1, CBF2, and CBF3 are rapidly and transiently upregulated in response to low temperature stress, typically reaching peak expression levels after exposure to 4°C for 1 to 3 hours. CBF can bind to downstream COR genes, and the proteins encoded by these COR genes can activate the expression of additional genes that encode osmotic regulators, either directly or indirectly. This process can increase the levels of soluble sugars and antifreeze proteins, repair membrane lipid damage caused by low temperatures, and ultimately improve the cold tolerance of the plants. Chinnusamy V et al. discovered that ICE1 specifically binds to the Avian myelocytomatosis virus (MYC) cis-acting element in the promoter region of the CBF3 gene. Furthermore, the overexpression of the ICE1 gene resulted in increased expression of the CBF3 gene and its downstream low-temperature-responsive genes in transgenic Arabidopsis thaliana plants subjected to low temperature stress. This enhancement significantly improved the cold tolerance of the transgenic plants.

In this experiment, we utilized the northern highbush blueberry variety ‘Duke’ as the test material to investigate the effects of various concentrations of EBR on several physiological indicators of cold resistance in blueberry buds, flowers, and young fruits. Additionally, we examined the expression of the key gene associated with cold resistance in blueberries, VcCBF3, under low temperature stress. The optimal concentration of exogenous EBR was identified to enhance the cold resistance of blueberries. This research aims to provide a cost-effective technology and methodology for improving cold resistance in blueberries during the flowering and young fruit stages. Furthermore, it seeks to establish a theoretical foundation for future studies on EBR's potential to mitigate damage caused by low temperature stress and to enhance the cold resistance of blueberries.”

Modification on the page 2-4, line 74-165.

Comments 6: 

Section 2.3.1 should be elaborated with detailed methods.

Response 6: 

Thank you for pointing this out. We agree with this comment.

We revise the section 2.3.1 to become “Pro content was determined using the ninhydrin color development method. A total of 0.5 g each of cut mixed buds, flowers, and young fruits were placed into stoppered test tubes. To each tube, 5 mL of a 3% sulfosalicylic acid solution was added. The tubes were then sealed and subjected to a boiling water bath for 10 min to facilitate extraction. After filtration, 2 mL of distilled water (control) and the filtrate were placed in separate test tubes. Subsequently, 2 mL of glacial acetic acid and 2 mL of ninhydrin reagent were added to each tube, which were then sealed and heated for 30 min in a boiling water bath. After cooling, 5 mL of toluene was added to each tube, and the mixture was shaken thoroughly to enhance extraction. The toluene layer was then carefully transferred to a cuvette. The samples were protected from light and allowed to stand until completely stratified. The absorbance of each sample was measured at 520 nm using a spectrophotometer, with the distilled water group serving as a blank control. Each measurement was repeated three times. The results were calculated using the following formula:

Where: C-mass of Pro concent from the standard curve, μg;

VT-total volume of extract, mL;

V1-volume of assay solution, mL;

W-sample mass, g.”

Modification on the page 6-7, line 239-258.

Comments 7: 

Section 2.4 - provides no details! poorly presented.

Response 7: 

Thank you for pointing this out. We agree with this comment.

We revise the section 2.4 to become “A one-way ANOVA and the Waller-Duncan test were employed to analyze the data. The data were imported into IBM SPSS version 26.0 to assess significant differences and were graphically represented using Origin 2022 and Microsoft Excel 2023 software.”

Modification on the page 13, line 478-491.

Dear Reviewer 2:

We are very grateful to Reviewer for reviewing the paper so carefully. We have tried our best to improve the manuscript and have modified. The revised paragraphs are labeled in green. Responds to the Reviewer’s comments were as follow:

Comments 1: 

In the introduction section, the authors should tell audiences more information. For example, how does blueberry deal with cold stress? how does EBR affect cold resistance in plant? Why can EBR be applied to blueberry? Additionally, there is a sentence in the first paragraph of introduction part is “The shrub is shallow-rooted and has weak cold resistance”. “Weak cold resistance” should be carefully used to blueberry.

Response 1: 

Thank you for pointing this out. We agree with this comment.

We revise the introduction to become “Blueberry, which belong to the rhododendron family (Ericaceae) and are classified under the genus Vaccinium, are perennial, shallow-rooted shrubs. They produce nutritious fruits that are both sweet and tart, accompanied by a delightful aroma. Changchun, located in northeastern China within an alpine region, experiences extended periods of low temperatures. In spring, as fruit trees begin to bud and flower, the flower organs exhibit a limited ability to withstand external damage. Temperature fluctuations, characterized by warming followed by cooling, can lead to occur ‘inverted spring cold’ phenomenon, resulting in the drying and wilting of flower buds, browning and dropping of flower organs, and even deformation of young fruits. Such conditions significantly hinder the development of the blueberry industry in the Changchun area. In a comparative study examining the cold resistance of various lingonberry varieties, Ren B concluded that the soluble sugar (SS) content increased as the temperature decreased. The soluble protein (SP) content exhibited an initial increase, followed by a subsequent decrease. Although the free proline (Pro) content increased, it gradually declined with the intensification of stress and the extension of the treatment duration. Furthermore, the malondialdehyde (MDA) content initially rose before slowly declining. Wei X et al. discovered that the MDA content in various blueberry branch varieties gradually increased as the temperature decreased, following an 'S'-shaped curve. In contrast, the activities of peroxidase (POD), superoxide dismutase (SOD), and catalase (CAT) exhibited a unimodal trend, initially increasing before subsequently decreasing. Liu BH et al. observed that the ascorbate-glutathione (AsA-GSH) system demonstrated a pattern of increase followed by a decrease as the temperature declined from the pre-dormant stage to the post-dormant stage.

In recent years, exogenous substances have attracted considerable attention for their role in enhancing plant cold resistance. 2,4-Epibrassinolide (EBR) is a novel class of highly effective, broad-spectrum, and environmentally friendly plant growth regulators. It has the capacity to regulate plant growth and development, improve resistance to low temperatures, and is recognized as the sixth major plant hormone on the global stage. EBR enhances the cold tolerance of plants by reducing the extent of membrane lipid peroxidation. It achieves this by increasing antioxidant capacity, lowering MDA levels, and decreasing the degree of lipid peroxidation in cellular membranes. Consequently, EBR improves the water-holding capacity of cells and tissues, thereby mitigating damage caused by low temperatures. Additionally, EBR modulates plant metabolism by inhibiting the generation of excessive free radicals and promoting the production

---

## [Editor Report · Decision Letter 1]

21 Oct 2024

Effects of exogenous EBR on the physiology of cold resistance and the expression of the VcCBF3 gene in blueberries during low-temperature stress

PONE-D-24-22975R1

Dear Dr. fan,

We’re pleased to inform you that your manuscript has been judged scientifically suitable for publication and will be formally accepted for publication once it meets all outstanding technical requirements.

Kind regards,

Mojtaba Kordrostami, Ph.D.

Academic Editor

PLOS ONE
---

## [Editor Report · Acceptance letter]

20 Nov 2024

PONE-D-24-22975R1 

PLOS ONE

Dear Dr. Fan, 

I'm pleased to inform you that your manuscript has been deemed suitable for publication in PLOS ONE. Congratulations! Your manuscript is now being handed over to our production team.

Kind regards, 

on behalf of

Dr. Mojtaba Kordrostami 

Academic Editor

PLOS ONE